# Microwave-Assisted Synthesis of High-Energy Faceted TiO_2_ Nanocrystals Derived from Exfoliated Porous Metatitanic Acid Nanosheets with Improved Photocatalytic and Photovoltaic Performance

**DOI:** 10.3390/ma12213614

**Published:** 2019-11-04

**Authors:** Yi-en Du, Xianjun Niu, Wanxi Li, Jing An, Yufang Liu, Yongqiang Chen, Pengfei Wang, Xiaojing Yang, Qi Feng

**Affiliations:** 1School of Chemistry & Chemical Engineering, Jinzhong University, Jinzhong 030619, China; xjniu1984@163.com (X.N.); liwanxi1986@163.com (W.L.); ann.jean@163.com (J.A.); sxyclyf@sina.com (Y.L.); 2Department of Advanced Materials Science, Faculty of Engineering, Kagawa University, 2217-20 Hayashi-cho, Takamatsu-shi 761-0396, Japan; 3Beijing Key Laboratory of Energy Conversion and Storage Materials, College of Chemistry, Beijing Normal University, Beijing 100875, China; 4State Key Laboratory of Coal Conversion, Institute of Coal Chemistry, Chinese Academy of Sciences, Taiyuan 030001, China; wangpf@sxicc.ac.cn

**Keywords:** anatase TiO_2_ nanocrystals, high-energy facets, photocatalytic activity, photovoltaic performance

## Abstract

A facile one-pot microwave-assisted hydrothermal synthesis of rutile TiO_2_ quadrangular prisms with dominant {110} facets, anatase TiO_2_ nanorods and square nanoprisms with co-exposed {101}/[111] facets, anatase TiO_2_ nanorhombuses with co-exposed {101}/{010} facets, and anatase TiO_2_ nanospindles with dominant {010} facets were reported through the use of exfoliated porous metatitanic acid nanosheets as a precursor. The nanostructures and the formation reaction mechanism of the obtained rutile and anatase TiO_2_ nanocrystals from the delaminated nanosheets were investigated. The transformation from the exfoliated metatitanic nanosheets with distorted hexagonal cavities to TiO_2_ nanocrystals involved a dissolution reaction of the nanosheets, nucleation of the primary [TiO_6_]^8−^ monomers, and the growth of rutile-type and anatase-type TiO_2_ nuclei during the microwave-assisted hydrothermal reaction. In addition, the photocatalytic activities of the as-prepared anatase nanocrystals were evaluated through the photocatalytic degradation of typical carcinogenic and mutagenic methyl orange (MO) under UV-light irradiation at a normal temperature and pressure. Furthermore, the dye-sensitized solar cell (DSSC) performance of the synthesized anatase TiO_2_ nanocrystals with various morphologies and crystal facets was also characterized. The {101}/[111]-faceted pH2.5-T175 nanocrystal showed the highest photocatalytic and photovoltaic performance compared to the other TiO_2_ samples, which could be attributed mainly to its minimum particle size and maximum specific surface area.

## 1. Introduction

Inorganic nanocrystals with tailored morphologies and specific facets have received much attention in the past decade due to their many intrinsic shape-dependent properties and excellent technological applications in energy and environmental fields [1,2]. As one of the most studied semiconductor metal oxides, titanium dioxide (TiO_2_) has been extensively utilized in photovoltaic cells, dye-sensitized soar cells, photocatalysis, the photocatalytic degradation of organic pollutants, Li-ion batteries, etc., due to its good photocatalytic activity, high biological and chemical inertness, long-term chemical and thermal stability, low price, nontoxicity, and excellent degradation capability [3,4,5,6,7]. Compared to the other two crystalline phases (rutile (tetragonal, space group *P*4_2_/*mnm*) and brookite (orthorhombic, space group *Pbca*)) of TiO_2_, anatase (tetragonal, space group *I*4_1_/*amd*) is widely accepted for its higher photocatalytic activity and for having the most superior dye-sensitized solar cell performance [8,9]. However, the photocatalytic and photovoltaic performances of anatase TiO_2_ nanocrystals still need to be further improved for their practical application and commercialization. To achieve this purpose, anatase TiO_2_ nanocrystals with high crystallinity, well-defined morphology, good architecture, a small crystallite size, a large specific surface area, and proper composition are desirable to improve photocatalytic and photovoltaic performance [10,11]. Besides, anatase TiO_2_ nanocrystals with well-defined morphologies and tailored high-energy crystal facets for photocatalysis and photoanodes also have been proven to be an effective approach to significantly improve photocatalytic and photovoltaic performance in recent years [12]. Surface scientists have demonstrated that the average surface energies of anatase TiO_2_ increase in the order of {111} facet (1.61 J/m^2^) > {110} facet (1.09 J/m^2^) > {001} facet (0.90 J/m^2^) > {010}/{100} facet (0.53 J/m^2^) > {101} facet (0.44 J/m^2^) [13,14]. Generally speaking, facets with high surface energies diminish rapidly during the crystal growth process for the minimization of the total surface free energy under equilibrium conditions, which results in the total exposed surface of the anatase TiO_2_ crystals being primarily controlled by {101} facets with poor reactivity and thermodynamic stability [15]. As a result, the morphology of the anatase TiO_2_ crystals is a slightly truncated tetragonal bipyramid with eight equivalent {101} facets and two equivalent {001} facets based on Wulff construction [8]. Therefore, it is imperative to reasonably design and synthesize various morphologies of TiO_2_ nanocrystals with exposed high-energy surfaces to enhance photocatalytic activity and optimize dye-sensitized solar cell (DSSC) performance. Important progress was made by Wen and coworkers, who reported the synthesis of nanosized anatase TiO_2_ crystallites with their dominant {010} facets exposed [16]. Subsequently, Yang and coworkers reported the synthesis of microsized anatase TiO_2_ crystals (with around 47% and 89% {001} facets being exposed) by using TiF_4_ as the raw material and hydrogen fluoride (HF) as the capping agent [15,17].Following these breakthroughs, more and more research work has focused on the design and control of the synthesis of anatase TiO_2_ crystals with varied percentages of high active facets [2,8,18,19,20]. However, the previously reported methods for synthesizing anatase TiO_2_ crystals with high-energy facets often involve the use of toxic and corrosive HF or other F-containing species, which restricts the practical application of TiO_2_ nanocrystals [10,12]. For example, Feng and coworkers reported the synthesis of regular foursquare anatase TiO_2_ mesocrystal sheets with dominant {001} facets by using TiCl_4_ as the titanium source and HF as the capping agent [21]. Illa and coworkers reported on hollow spheres assembled by high-energy {001}-faceted mesoporous cuboid anatase nanocrystals, which showed higher photocatalytic and photoelectrochemical activity than did those of commercial P25 [22]. Khalil and coworkers reported the synthesis of Au–TiO_2_ heterostructures with exposed {001} facets, which showed enhanced photocatalytic performance [23]. Ye and coworkers reported the synthesis of anatase TiO_2_ crystals with co-exposed {101}/{001} and {010}/{001} facets by using (NH_4_)_2_TiF_6_ as the titanium and fluorine species [24]. Xu and coworkers reported the synthesis of square-shaped plate TiO_2_ single crystals with well-defined {111} facets that were exposed using both HF and ammonia as the capping reagents [14]. Amoli and coworkers reported the synthesis of anatase TiO_2_ nanocubes and nanoparallelepipeds with well-defined {111} facets exposed by using oleylamine as the morphology controlling agent and NH_3_/HF as the stabilizing agent [25]. Recently, we also synthesized truncated tetragonal bipyramid anatase TiO_2_ nanocrystals with co-exposed {001}, {010}, and {101} facets and tetragonal cuboid anatase TiO_2_ nanocrystals with co-exposed [111]- and {101} facets by using the H^+^ form of tetratitanate H_2_Ti_4_O_9_ as the precursor and HF as the capping agent [26]. Very recently, using porous metatitanic acid H_2_TiO_3_ as the precursor and HF as the capping agent, cuboid-like anatase TiO_2_ nanocrystals with co-exposed {101}/[111] facets and irregular anatase TiO_2_ nanocrystals with co-exposed {101}/{010} facets were also synthesized [27].

The microwave-assisted hydrothermal method is unique in its ability to be scaledup without suffering thermal gradient effects, providing a potentially industrially important improvement over convective methods in the synthesis of nanocrystals [28]. The advantage of using the microwave-assisted hydrothermal method over a conventional heating method lies in its homogenous selective heating and scalability, its shorter reaction time, its rapid formation rate, and the higher purity and better crystallinity of the produced products (as the yields obtained are identical for both heating methods) [29,30,31]. For example, under microwave-assisted solvothermal conditions, Yoon and coworkers reported the synthesis of various polymorphs of TiO_2_ nanocrystals with different morphologies by using TiCl_4_ or TiCl_3_ as precursors [29]. Nunes and coworkers reported the production and photocatalytic activity of brookite/rutile mixed-phase TiO_2_ nanorod spheres and nanorod arrays grown on polyethylene terephthalate substrates, which displayed remarkable degradability performance and reusability under ultraviolet and solar radiation for the degradation of rhodamine B [32].

In the present work, we report a facile synthesis of shape-tunable TiO_2_ nanocrystals with dominant {110}, {010}, and [111] facets through a simple one-step fluorine-free microwave-assisted hydrothermal method. The synthetic process demonstrates that the pH value of the exfoliated nanosheet solution plays a key role in morphology evolution and facet exposure under hydrothermal conditions. The possible transformation reaction mechanism of the TiO_2_ nanocrystals with various morphologies and exposure facets and the photocatalytic and photovoltaic performance were investigated. Compared to P25 TiO_2_ nanocrystals, the pH2.5-T175 nanocrystals with dominant [111] facets showed higher photocatalytic and photovoltaic performance.

## 2. Materials and Methods 

### 2.1. Preparation of Layered Li_2_TiO_3_and H_2_TiO_3_

Lithium metatitanate (Li_2_TiO_3_) was formed through a conventional solid-state reaction [33]. Lithium carbonate (Li_2_CO_3_, 99.0%) and titanium dioxide (TiO_2_, 98.5%) were taken in a molar ratio of 1.05:3 and milled in a horizontal ball mill at 50 rpm for 6 h, and then the mixture was calcined at 850 °C for 24 h in an air atmosphere. Then, 10.0 g of the obtained earthy gray Li_2_TiO_3_ powder was acid-treated three times with a 1.0 mol·L^−1^ HCl solution (1 L) at room temperature for 24 h under magnetic stirring to remove Li^+^ completely from the metatitanate to obtain a white metatitanic acid H_2_TiO_3_ sample.

### 2.2. Exfoliation of H_2_TiO_3_ into a Nanosheet Colloidal Solution

The obtained metatitanic acid H_2_TiO_3_ sample (5.0 g) was hydrothermally treated at 100 °C for 24 h with magnetic stirring in a 12.5% tetramethylammonium hydroxide solution (50 mL, (CH_3_)_4_NOH, hereafter TMAOH) to intercalate TMA^+^ ions into the interlayer of the metatitanic acid H_2_TiO_3_ and to obtain a TMA^+^-intercalated layered H_2_TiO_3_ sample. The intercalation compound obtained was delaminated into H_2_TiO_3_ nanosheet by dispersed it in 450 mL of deionized water with stirring at room temperature for 3 days. 

### 2.3. Microwave Hydrothermal Synthesis of TiO_2_ Nanocrystals 

TiO_2_ nanocrystals were prepared through microwave-assisted hydrothermal treatment of the above exfoliated H_2_TiO_3_ nanosheet colloidal solution (40 mL) at a desired temperature for 2 h. After the microwave hydrothermal process, the resultant sample was filtered, washed multiple times with deionized water, and then dried using a freeze-drying machine. Here, the obtained TiO_2_ sample is specified as pH*x*-T*y*, where *x* and *y* are the desired pH value of the nanosheet colloidal solution and the desired temperature of the microwave-assisted hydrothermal treatment, respectively.

### 2.4. Photocatalytic Activity 

The photocatalytic activity of the resulting TiO_2_ nanocrystals was assessed through the photodecomposition of carcinogenic and mutagenic methyl orange (MO), one of the most stable azo dyes [34,35]. Here, 200 mL of 2.31 × 10^−5^ mol/L MO solution was added to a 250mL quartz glass beaker containing 75 mg of as-prepared TiO_2_ nanocrystals and then sonicated for 5 min for the complete dispersion of the TiO_2_ nanocrystals followed by continuous stirring for 60 min in the dark to prepare the suspension solution of TiO_2_ nanocrystals. The above suspension solution was placed in the dark for 2 days to ensure that the adsorption–desorption phenomenon reached equilibrium. After that, photocatalytic degradation was carried out by stirring the above suspensions under ultraviolet irradiation using a 175-W low-pressure mercury lamp (λ = 365 nm). The distance between the mercury lamp and the suspension solution was 30 cm. At intervals of 15 min, 4 mL of suspension solution was taken out from the beaker and centrifuged at 2000 rpm for 5 min to separate the TiO_2_ nanocrystals, and the concentration of MO supernatant solution was analyzed by using a TU-1901 spectrophotometer (Beijing Purkinje General Instrument Co. Ltd. Beijing, China). The photodegradation efficiency (*η*) of the as-prepared TiO_2_ nanocrystals (as a percentage) was calculated according to the equation [36]:
$$\eta =\frac{c_{0}-c_{t}}{c_{t}}\times 100\%$$
where *c*_0_ and *c*_t_ represent the concentration of the MO suspensionin the absence and presence of light irradiation, respectively. For a comparison, Degussa P25-TiO_2_ (~87% anatase and ~13% rutile) was also investigated. 

### 2.5. Fabrication of Photoanodes and Dye-Sensitized Solar Cells (DSSCs)

For the preparation of the nanoporous TiO_2_ layers with the prepared TiO_2_ nanocrystals and the Degussa P25-TiO_2_ nanocrystals, a viscous slurry was prepared through the following procedure. The prepared TiO_2_ nanocrystals (0.5 g) were dispersed in ethanol (2.5 g) and then mixed with α-terpineol (2.0 g), ethyl-cellulose 10 (1.4 g of 10 wt% solution in ethanol), and ethyl-cellulose 45 (1.1 g of 10 wt% solution in ethanol). Subsequently, the resulting mixture was sonicated for 5 min and then ball-milled for 3 days. After removing the ethanol with a rotary evaporator, an optical paste containing 18% TiO_2_, 9% ethyl-cellulose, and 73% α-terpineol was obtained. A nanoporous TiO_2_ photoanode was obtained according to the following steps. Fluorine-doped tin oxide (FTO)-conducting glass substrates were cleaned in the order of neutral cleaner, deionized water, and absolute ethanol by ultrasonication for 5 min. Subsequently, the FTO glass substrates (12.5 × 25.0 mm^2^) were dipped into titanium tetraisopropoxide solution (0.1 mol·L^−1^) for 1 min and then washed with ionized water and absolute ethanol. After being dried, the prepared FTO glass substrates were placed in a muffle furnace, and the reaction temperature was increased to 480 °C at 10 °C/min and was kept at this temperature for 1 h to coat a dense TiO_2_ film on the surface of the FTO glass substrates. Afterwards, the prepared TiO_2_ paste was coated on the above FTO glass substrates by using the doctor-blade technique, was dried at 100 °C for 15 min, and was sintered in the above muffle furnace at 315 °C for 15 min. This process was repeated several times until the desired thickness of TiO_2_ film was obtained. The above FTO glass substrates coated with the TiO_2_ pastes were first sintered at 450 °C for 30 min, then treated with 0.1 mol·L^−1^ titanium tetraisopropoxide solution as described above, and finally sintered at 480 °C for 1 h to fabricate TiO_2_ thin-film photoanodes. After being cooled to 80 °C, the TiO_2_ photoanodes were immersed in N719 dye solution for 24 h at room temperature. An N719-sensitized TiO_2_ photoelectrode (working electrode) was coupled with a platinum-sputtered FTO glass substrate (counter-electrode) with an electrolyte solution between the electrodes to assemble a sandwich-type DSSC. The electrolyte of the DSSC consisted of a solution of 0.6 mol·L^−1^ 1-butyl-3-methylimidazolium iodide, 0.03 mol·L^−1^ I_2_, 0.1 mol·L^−1^ guanidinium thiocyanate, and 0.5 mol·L^−1^ tert-butylpyridine in a cosolvent of acetonitrile and valeronitrile at a volume ratio of 85:15, which was injected into the cell through capillary forces.

### 2.6. Characterization

X-ray diffraction (XRD) patterns of the resulting specimens were prepared by a XRD-6100 powder X-ray diffractometer (SHIMADZU, Kyoto, Japan) with Cu Kα radiation. Field-emission scanning electron microscopy (FESEM, HITACHI, Tokyo, Japan) and a transmission electron microscope (TEM, JEOL, Tokyo, Japan) were used to characterize the morphological and structural properties of the TiO_2_ nanocrystal specimens. The specific surface area of the specimens was determined through aBrunauer-Emmett-Teller (BET) analysis (Quantachrome Quadra Win Instruments, Ashland, VA, USA) using N_2_ as the adsorbate gas at −196 °C. All of the specimens were degassed at 120 °C for 3 h prior to nitrogen adsorption measurements. Photovoltaic DSSC performance and photocurrent–voltage characteristic curves were measured using a Hokuto-Denko BAS100B electrochemical analyzer (Hokuto, Yamanashi-ken, Osaka, Japan). The effective irradiation area of the DSSC was fixed at 0.25 cm^2^.

## 3. Results and Discussion

### 3.1. Preparation of the Metatitanic Acid Titanate Nanosheet Colloidal Solution

The XRD pattern of the layered Li_2_TiO_3_ sample obtained at 850 °C for 24 h in an air atmosphere is shown in Figure 1a. The diffraction peaks recorded at 2θ = 18.59°, 20.40°, 36.10°, 43.78°, 47.86°, 57.78°, 63.46°, and 66.90° can be attributed to the reflections of (002), (020), (−131), (−133), (−204), (006), (−206), and (062) crystal planes of layered Li_2_TiO_3_, which were almost in accordance with the characteristic diffraction peaks of the monoclinic Li_2_TiO_3_ crystal (*C*2/*c* space group, JCPDS No. 33-0831). The interlayer spacing was estimated to be 4.47 Å according to the (002) crystal plane of Li_2_TiO_3_, and the chemical formula (Li_2.06_TiO_3.0_) (on the basis of the chemical analyses) was close to the theoretical formula of Li_2_TiO_3_. The corresponding crystal structure of layered Li_2_TiO_3_ is shown in Figure 1d. The crystal structure of Li_2_TiO_3_ could be represented as a cubic close packing of three different layers (Li_6_ layers, O_6_ layers, and Li_2_Ti_4_ layers) in the *c* axis direction [37]. In the Li_2_Ti_4_ layer, 1/3 of the 4e sites were occupied by lithium atoms and 2/3 by titanium atoms, so the formula for a well-ordered Li_2_TiO_3_ could be written as Li(Li_1/3_Ti_2/3_)O_2_ [37,38,39]. That is, the atomic fraction of titanium in the Li_6_ layers and Li_2_Ti_4_ layers was 75% and 25%, respectively [38]. Lithium extraction in the layered Li_2_TiO_3_ precursor was performed using 1 mol·L^−1^ hydrochloric acid. According to the chemical analyses (H_1.993_Li_0.007_TiO_3_), the amount of lithium extracted was about 99.65%. Figure 1b depicts the XRD pattern of H_2_TiO_3_. The main peak corresponding to (002) of H_2_TiO_3_ slightly shifted to a higher 2θ angle compared to the peak in Figure 1a, indicating a decrease in the interlayer (from 4.77 to 4.72 Å) through an exchange of the lithium atoms (radius of ~0.076 nm) for hydrogen ions (radius of ~0.0012 nm) [33]. It can be seen that the XRD pattern in Figure 1b is similar to that in Figure 1a, which suggests that the structure did not change much after lithium extraction. The FESEM image of Li_2_TiO_3_ shows that the irregular particles with smooth surfaces were nonagglomerated, and the average size of the particles was in the range of 0.5–2.0 μm (Figure 2a). The particle shape of H_2_TiO_3_ (nonagglomerated) was similar to that of the Li_2_TiO_3_ precursor, except that the smooth surface of the H_2_TiO_3_ particles became rough and irregular cracks appeared on the surface (Figure 2b,c), implying that the extraction process hardly destroyed the structure of the precursor even if the Li^+^ ions in the Li_2_Ti_4_ intralayers were extracted. The XRD pattern of the TMA^+^-intercalated H_2_TiO_3_ sample (i.e., the dried TMA^+^-exfoliated H_2_TiO_3_ nanosheets) is shown in Figure 1c. Compared to the diffraction peaks of the H_2_TiO_3_, a set of new diffraction peaks was observed at ~5.30° (002), ~10.52° (004), and ~15.84° (006) in the TMA^+^-intercalated H_2_TiO_3_ sample. The basal diffraction peak of H_2_TiO_3_ significantly shifted to a lower 2θ angle (Figure 1b), indicating the expansion of the interlayer (from 4.72 to 16.66 Å) through the intercalation of (CH_3_)_4_N^+^. The gallery heights of [(CH_3_)_4_N]_2_TiO_3_, which were determined by subtracting the TiO_3_^2−^ layer thicknesses of 4.6 Å, were 12.06 Å (16.66–4.6 Å). Since the height of (CH_3_)_4_N^+^ and the diameter of a water molecule are ca. 5.3 and 2.8 Å [40,41], respectively, it was suspected that two molecules of water (2 × 2.8 Å) and a molecule of (CH_3_)_4_N^+^ (5.3 Å) were vertically arranged in the interlayer. Overall, the data in Figure 1c shows that [(CH_3_)_4_N]_2_TiO_3_ formed nanosheet stacks after drying [42].

### 3.2. Synthesis of TiO_2_ Nanocrystals from Metatitanic Acid Nanosheet Colloidal Solution

TiO_2_ nanocrystals with different structures and morphologies were rapidly synthesized by microwave-assisted hydrothermal treatment of the prepared metatitanic acid nanosheet colloidal solution at various pH values (0.5–13.5) and temperatures (105–185 °C) for 2 h. The crystallographic structures of the TiO_2_ nanocrystals obtained at different pH values and various temperatures were identified by XRD analysis (see Appendix A). The dependence of the TiO_2_ nanocrystals on the microwave-assisted hydrothermal reaction conditions is summarized in Figure 3. It can be seen that the rutile phase was formed preferentially under strong acidic conditions (pH ≤ 0.5). The mixed-phases of rutile and anatase were formed by the microwave-assisted hydrothermal metatitanic acid nanosheet colloidal solution within a certain pH range (0.5 < pH ≤ 1.0). The rutile phase disappeared when the pH increased to 1.5, indicating that the rutile phase was stable under strong acidic conditions and that its stability was greater than anatase. A pure anatase phase could be obtained in a wide pH range of 1.5 ≤ pH ≤ 13.5 at a certain temperature, and its crystallinity increased with increasing temperature and pH values (see Appendix A). The unreacted layered compound was observed at pH ≥6.5 and at low temperatures (see Appendix A) and was formed through the restacking of the exfoliated metatitanic acid nanosheets, implying that higher pH values and lower temperatures are not conductive to a transformation from nanosheets to anatase nanocrystals.

Figure 4 shows the XRD patterns of the typical samples at different pH values varying from 0.5 to 13.5. Apparently, for the pH0.5-T175 sample, the diffraction peaks (2θ values) of 27.42°, 36.06°, 38.29°, 41.24°, 44.12°, 54.30°, 56.56°, 62.82°, 64.00°, and 69.04°, could be assigned to (110), (101), (200), (111), (210), (211), (220), (002), (310), and (301) crystalline surfaces of the rutile TiO_2_ phase, respectively (JCPDS No. 21-1276, Figure 4a). With an increasing pH value in the suspension liquid, the pH1.0-T175 sample exhibited an anatase/rutile coexisting diffraction pattern, as shown in Figure 4b. The peaks at 2θ values of 27.40°, 36.04°, 41.28°, 54.34°, 56.70°, 62.88°, and 69.05° could be ascribed to (110), (101), (111), (211), (220), (002), and (301) crystalline surfaces of the rutile phase, respectively (JCPDS No. 21-1276), while other diffraction peaks at 25.46°, 37.82°, and 48.02° arose from the (101), (004), and (200) crystalline surfaces of the anatase TiO_2_ phase, respectively (JCPDS No. 21-1272). The mass fraction of anatase and the rutile contents present in the sample could be accurately obtained from the following equations [43]:(1)$$W_{A}=\frac{1}{1+1.26[I_{R}(110)/I_{A}(101)]}$$
(2)$$W_{R}=\frac{1}{1+0.8[I_{A}(101)/I_{R}(110)]}$$
where *W*_A_ and *W*_R_ represent the mass fraction of anatase and rutile in the mixed phase, respectively, and *I*_A_(101) and *I*_R_(110) represent the diffraction peak integral intensity of the anatase TiO_2_ (101) crystal surface and the rutile TiO_2_(110) crystal surface, respectively. As shown in Figure 4b, the diffraction peak integral intensity of the anatase TiO_2_ (101) crystal surface and rutile TiO_2_ (110) crystal surface was 60.7% and 100.0% in the mixed phase, respectively. Therefore, the mass fraction was 32.51% for anatase TiO_2_ and 67.31% for rutile TiO_2_. With the pH values further increasing to 1.5, the rutile phase decreased to almost null, which means that the H_2_TiO_3_ nanosheets were completely transformed to anatase nanocrystals and that the entire structure became anatase (Figure 4c–j), i.e, the higher pH inhibited the growth of rutile TiO_2_ and promoted the growth of anatase TiO_2_ during the microwave-assisted hydrothermal reaction process. The various diffraction peaks observed at ~25.66° (101), ~38.22° (004), ~48.32° (200), ~54.10° (105), ~54.50° (211), ~62.86° (204), and ~68.67° (116) were the characteristic peaks of anatase TiO_2_ (JCPDS No. 21-1272). It could be found that the broadening diffraction peaks became sharper with increasing pH values, demonstrating that the grain size of the anatase matrix increased with increasing pH values. The increase in grain size was attributed to the Ostwald ripening phenomena during the particle growth process [43].

### 3.3. Morphology and Exposed Crystal Facets of TiO_2_ Nanocrystals 

Figure 5a depicts a TEM image of the pH1.5-T175 nanocrystals derived from microwave-assisted hydrothermal treatment of the exfoliated H_2_TiO_3_ nanosheet colloidal solution at 175 °C with a reaction time of 2 h. As shown, a large number of rod-shaped nanocrystals with a size of about 10–30 nm and a small number of rhombic nanocrystals with a size of ~10 nm were observed. The corresponding high-resolution TEM (HRTEM) images (Figure 5b,c) showed unparallel (101) and (011) atomic planes of the rod-shaped nanocrystals with a lattice spacing of 3.44 and 3.47 Å (or 3.42 Å), respectively, and an interfacial angle of 82°, which matches well with the theoretical value for the angle between the {101} and {011} facets of anatase TiO_2_ nanocrystals [14]. Therefore, the dominant exposed crystal plane of the pH1.5-T175 nanocrystals was perpendicular to the above {101} and {011} crystal facets, i.e., the crystal plane was vertical to the [111] crystal zone axis. Since the crystal plane perpendicular to the {101} and {011} crystal facets was uncertain, for convenience, we express it as an [111] facet. The exposed rod-shaped nanocrystal [111] crystal facet could be further confirmed through a fast-Fourier-transform (FFT) diffraction pattern (Figure 5c inset). It should be noted that the [111] facet was different from the {111} facet, because anatase belongs to a tetragonal system, not a cubic system. In a cubic crystal system, the crystal plane perpendicular to [111] the crystal zone axis is the {111} facet. According to the above analysis, the rod-shaped nanocrystals co-exposed in the crystal planes were [111] facets and {101} facets, i.e., {101}/[111] facets. Figure 5d shows the unparallel (101) and (002) planes of the rhombic nanocrystals with a lattice spacing of 3.38 and 4.62 Å, respectively. Moreover, the interfacial angle of ca. 68.3° between the aforementioned atomic planes also matches well with the theoretical value of anatase TiO_2_ nanocrystals [15,44]. Therefore, the above rhombic nanocrystal was confirmed to be anatase TiO_2_ with dominant {010} facets on the two basal surfaces and {101} facets on the four lateral surfaces. Figure 5e–h presents HRTEM images of pH2.5-T175. As shown, nanocrystals with short rod-shaped, square-prism-shaped, and some irregular morphology were observed. Along the [111] zone axis, the HRTEM image in Figure 5e,f,h displays (101) and (011) crystal facets of the rod-shaped (or square-prism-shaped) nanocrystal with an interfacial angle of 82°, which is in good agreement with the theoretic value. Therefore, the above rod-shaped and square-prism-shaped anatase TiO_2_ nanocrystals co-exposed {101}/[111] facets. Furthermore, lattice fringes with a spacing of 3.50 Å could be assigned to the {101} facets of the irregularly shaped anatase nanoparticles. The FFT diffraction pattern of the yellow dashed lines region (Figure 5e,g,h inset) further indicated that the rod-shaped, square-prism-shaped, and irregular TiO_2_ nanocrystals were single-crystalline.

As shown in Figure 6a,b, the synthesized pH4.5-T175 nanocrystals (anatase phase) mainly had two morphologies, a square prism and a rhombus. Moreover, a small number of anatase nanocrystals with irregular morphologies were also observed in the above sample. Figure 6c,d shows an HRTEM image of typical square-prism-shaped anatase TiO_2_ nanocrystals. The lattice fringes of 3.52 and 3.52 Å corresponded to {101} and {011} lattice spacing, respectively, and the angle of 82° measured between the {101} and {011} facets matched closely with the theoretical value, corroborating that the exposed crystal facets were [111] facets and {101} facets (or {011} facets) on the two basal surfaces and the four lateral surfaces of the square-prism-shaped anatase TiO_2_ nanocrystals, respectively. According to the above discussion, the irregular anatase TiO_2_ nanocrystals in Figure 6e co-exposing the crystal facets were also {101}/[111] facets. The lattice fringes with spacing of 3.51 Å could be assigned to the {101} facet of the approximately rhombic anatase TiO_2_ nanocrystal (Figure 6f), which was parallel to the lateral surface, indicating that the lateral surface had exposed {101} facets. 

Figure 7a,c shows a TEM image of the product obtained at pH = 6.5 and *t* = 175 °C with a reaction time of 2 h. As shown, a large number of well-defined spindle-shaped nanocrystals 70–130 nm in length and 15–30 nm widewere observed. In addition, some square-prism-shaped nanocrystals 20–50 nm in length and 20–30 nm wide were also observed. Figure 7b,d includes corresponding HRTEM images taken from the marked area of the TEM images in Figure 7a,c, respectively. As shownin Figure 7b, the spindle-shaped nanocrystal displayed the (101), (10−1), and (002) atomic planes with lattice spacings of 3.53, 3.53 and 4.76 Å, and the angles *α*, *β*, *γ* between the (101) and (002), (10−1) and (002), and (101) and (10−1) facets were 68.3°, 68.3°, and 43.4°, respectively, the same as the theoretical values [45]. These facts prove that the spindle-shaped anatase nanocrystal exhibited four flat facets {010} facets on the four vertical surfaces, eight inclined {101} facets on the slant surfaces, and two parallel {001} facets on the top and bottom surfaces. Along the [010] zone axis, the HRTEM image in Figure 7d also displays (101) and (002) crystal facets with an interfacial angle of 68.3°, indicating a spindle-shaped anatase nanocrystal with exposed {010} facets on the four vertical surfaces. 

The well-defined nanospindle structure (with a particle size of about 70–140 × 17–35 nm^2^) of the pH8.5-T175 sample was further confirmed by the TEM image, as shown in Figure 8a. Two magnified TEM images of the uniform anatase nanospindles are presented in Figure 8c,e. Figure 8b,d,f shows HRTEM images marked from the yellow dotted frame area indicated in Figure 8a,c,e, respectively. There were three types of lattice with fringe spacing of 3.53, 4.74, and 3.53 Å (or 3.49, 4.78, and 3.49 Å) and interfacial angles of 68.3°, 68.3°, and 43.4°, which agreed well with the (101), (002), and (10−1) lattice planes of anatase TiO_2_. The {001} facets were perpendicular to the long axis (*c* axis) direction and parallel to the transverse axis (*b* axis) direction of the spindle-shaped nanocrystal, respectively, indicating the oriented growth of spindle-shaped nanocrystals in the [001] crystallographic direction. Therefore, the exposed lattice facets of the nanospindles were mainly {010} facets (based on the HRTEM analysis).

Figure 9 displays typical FESEM images of the samples derived from microwave-assisted hydrothermal treatment of the exfoliated H_2_TiO_3_ nanosheet colloidal solution at various pH values (0.5–2.5) and temperatures (155–185 °C). The pH0.5-T*x* (*x =* 155, 165, 175, or 185) sample exhibited quadrangular prism morphology with a length of 50–100 nm and a width (or thickness) of 15–30 nm, which corresponded with the rutile TiO_2_ phase and the elongation direction of the nanocrystals along the [001] crystallographic direction (Figure 9a–d) [46]. The average size of the quadrangular prism nanocrystals did not change obviously with an increase in the temperature, indicating that an increase in temperature mainly increased the crystallinity of the nanocrystals during the microwave-assisted hydrothermal reaction, which was consistent with the results of the XRD analysis. As shown in Appendix A, the intensity of the diffraction peaks of the rutile phase at 2θ = 27.42° (110), 36.06 (101), 38.29° (200), 41.24° (111), 54.30° (211), and 56.56° (220) increased with an increase in the reaction temperature (from 105 to 185 °C), indicating that the crystallinity of rutile nanocrystals improved. For the pH1.0-T175 sample, some bigger rod-shaped nanocrystals (rutile phase) and lots of smaller nanocrystals (anatase phase) with various morphologies were observed (Figure 9e). With a further increase in the pH value, rod-shaped (or square-prism-shaped) nanocrystals with a size of about several tens of nanometers were observed, all of them belonging to the anatase phase (Figure 9f–i). The results showed that the pH value of the nanosheet colloidal solution had a great influence on the crystal structure and morphology of the produced TiO_2_. 

As shown in Figure 10a–c, the pH4.5-T*y* (*y* = 165, 175, or 185) sample mainly exhibited square-prism and rhombus morphologies with a particle size of ~15–30 nm. With the increase of temperature, the particle size did not increase significantly, which indicated that the increase of temperature only increased the crystallinity of the particles, which was consistent with the results of the XRD analysis. As shown in Appendix A, the intensity of the characteristic diffraction peaks of the anatase phase at 2θ = 25.66° (101), 38.22° (004), 48.32° (200), 54.10° (105), 54.50° (211), and 62.86° (204) increased with increases in the reaction temperature (from 115 to 185 °C), indicating that the crystallinity of anatase nanocrystals increased with an increase in the reaction temperature. Figure 10d–j presents the FE-SEM images of pH*x*-T*y* (*x* = 6.5, 8.5, 10.5, or 12.5; *y* =125, 165, 175, or 185) products, which revealed that the obtained TiO_2_ products consisted of uniform, well-defined, spindle-shaped structures with an axial length of 75–155 nm and a transverse width of 20–35 nm. Interestingly, a series of directional streaks could be observed on the surface of the spindle-shaped crystal, which were perpendicular to the long axis (*c* axis)direction and parallel to the transverse axis (*b* axis) direction of the spindle-shaped nanocrystals, indicating that the orientation growth was in the [001] crystallographic direction. The crystal size increased with an increase in the pH value (from 6.5 to 12.5), especially for the [001] orientation of the spindle-shaped nanocrystals. Arrow-shaped crystals 500–700 nm in length were formed at pH13.5, as shown in Figure 10l. Spindle-shaped, arrow-shaped nanocrystals and unreacted nanosheets were observed at lower temperatures, indicating that a lower temperature was not conducive to the fracturing of the nanosheet (Figure 10g,k).

### 3.4. Transformation Reaction from the Delaminated H_2_TiO_3_ Nanosheets to TO_2_ Nanocrystals 

The nanocrystal conversion process between the exfoliated H_2_TiO_3_ nanosheets and the TiO_2_ nanocrystals can be described as the schematic diagram in Figure 11. On the basis of the discussion above, the transformation reaction mechanism from the exfoliated metatitanic nanosheets with distorted hexagonal cavities to TiO_2_ nanocrystals can be described by the following chemical equations:[TiO_3_]^2−^ + 3H_2_O → [TiO_6_]^8−^ + 6H^+^(3)
[TiO_6_]^8−^ + 8H^+^ → TiO_2_ + 4H_2_O(4)
[TiO_6_]^8−^ + 4H_2_O → TiO_2_ + 8OH^−^(5)

First, the delaminated [TiO_3_]^2−^ nanosheets were split into many primary [TiO_6_]^8−^ monomers along the edge-shared oxygen atoms under microwave-assisted hydrothermal conditions. At this stage, the pH values of the [TiO_3_]^2−^ nanosheet colloidal solution had an important influence on the reaction (3). As shown in Figure 3, the acidic conditions were advantageous to the reaction (Equation (3)), which could be carried out at relatively lower temperatures, while the alkaline conditions were disadvantageous to the reaction (Equation (3)), which needed to be carried out at relatively higher temperatures. Furthermore, the rutile-type and anatase-type TiO_2_ nuclei were formed by polymeric [TiO_6_]^8−^ monomers along the equatorial and apical edges at different pH values, respectively. Rutile nuclei could be formed under super acidic conditions (Equation (4)), and anatase nuclei could be formed under weak acidic, neutral, or alkaline conditions (Equations (4) and (5)). Finally, the different types of TiO_2_ nanocrystals with various morphologies and exposed facets were formed through the directional growth of the TiO_2_ nuclei.

The growth of the rutile-type and anatase-type TiO_2_ nuclei along the different crystallographic directions caused the formation of a rutile quadrangular prism, an anatase nanorod, an anatase nanorhombus, an anatase square nanoprism, and an anatase nanospindle. As shown in Figure 11, a rutile quadrangular prism with exposed {110} facets on the four lateral surfaces was formed by the directional growth of the rutile-type TiO_2_ nuclei along the [001] crystallographic direction under strong acidic conditions (pH = 0.5, 1.0). An anatase nanorod (or square nanoprism) with co-exposed {101}/[111] facets was formed by the directional growth of the anatase-type TiO_2_ nuclei along the [101] and [011] crystallographic directions, and an anatase nanorhombus with dominant exposed {010} facets was formed by the directional growth of the nuclei of anatase-type TiO_2_ along the [101] and [001] crystallographic directions at relatively low pH values (1.0 ≤ pH ≤ 4.5). At pH ≥ 6.5, an anatase nanospindle with dominant exposed {010} facets was formed by the directional growth of the nuclei of anatase-type TiO_2_ along the [001] crystallographic direction.

### 3.5. Photocatalytic Activities of the As-Synthesized TiO_2_ Nanocrystals 

The photocatalytic activities of the TiO_2_ nanocrystal samples synthesized in the present work were quantitatively evaluated via bleaching 7.5 mg/L MO solutions under UV light irradiation at room temperature. The photocatalytic degradation of MO can be considered as the following reactions [47,48]: $${\mathrm{TiO}}_{2}+hv\to {\mathrm{TiO}}_{2}(e^{-}+h^{+})\;h^{+}+{\mathrm{OH}}^{-}\to \cdot \mathrm{OH}\;e^{-}+O_{2}\to O_{2}{}^{-}\;O_{2}{}^{-}+H_{2}O\to {\mathrm{HO}}_{2}\cdot +{\mathrm{OH}}^{-}\;{\mathrm{HO}}_{2}\cdot +H_{2}O\to H_{2}O_{2}+\cdot \mathrm{OH}\;H_{2}O_{2}\to 2\cdot \mathrm{OH}$$

·OH + MO → peroxides or hydroxylated intermediates→ degraded or mineralized products.

The above reactions were selected because both the photoelectrons and photoholes eventually participate in the formation of the OH radical, which degrades MO. In order to have a better evaluation of the photocatalytic efficiency of the as-prepared ^−^TiO_2_ nanocrystals, Degussa P25 (average particle size of 26.2 nm) was chosen as the photocatalytic reference. Figure 12a and Appendix A show the UV-Vis absorption spectra of the centrifuged MO solutions at certain intervals during the photodegradation treatment with and without TiO_2_ nanocrystals. As can be seen, the intensity of the peaks in the visible region gradually decreased with an extension of the irradiation time. Moreover, it can be observed that the strongest peaks at 464 nm shifted toward the shorter wavelength direction (i.e., a hypsochromic shift), which can be attributed to the demethylation of the MO [49]. Demethylation and cleavage of the MO chromophore ring structure occurred simultaneously during the initial period of photodegradation of MO, in which demethylation played a predominant role [49]. With the prolongation of UV light irradiation time, the demethylated MO intermediates could be further decomposed, which was indicated by the change in peak intensity at 435 nm (Figure 12a). However, the intensity of the peaks changed little after 90 min of UV irradiationin the absence of the TiO_2_ sample (see Appendix A), which indicated that the bleaching/degradation of the MO aqueous solution occurred on the surface of the TiO_2_ photocatalyst [35]. 

Figure 12b shows the evolution of the photodegradation rate along with the irradiation time. It can be noticed that the pH2.5-T175 nanocrystal showed the highest decomposition efficiency among all the tested samples, reaching above 96.5% under UV irradiation for 90 min. The second fastest decomposition efficiency was obtained with the pH4.5-T175 nanocrystal, where 93.3% of MO was decomposed after 90 min of UV light irradiation. The order of decomposition efficiency observed for MO degradation in the other samples was P25 (81.1%) > pH6.5-T175 (44.7%) > blank sample (3.1%). In order to have a quantitative comparison of the decomposition efficiency of the above samples, the experimental data in Figure 12b were fitted according to the pseudo-first-order kinetic process, which can be expressed as follows [50]:
ln(*c*_0_/*c*_t_ ) = *kt*
where *k* is the photocatalytic degradation rate constant (min^−1^), and *t* is the degradation time. The fitted curves are shown in Figure 12c, and the obtained *k* (min^−1^), as well as the correlative coefficient (*R*^2^) of all the curves, is listed in Table 1. Obviously, the curves agreed well with the experimental data points. It can be seen that the pH2.5-T175 nanocrystal showed the highest *k* value of 0.0346 min^−1^, almost 1.23, 1.85, 5.32, and 86.5 times as high as that of pH4.5-T175, P25, pH6.5-T175, and the blank sample (Table 1), respectively, which helped us to have a better understanding of the pH2.5-T175 nanocrystal having the highest performance among all of the tested samples.

It is well known that the photocatalytic activity of a photocatalyst can be strongly influenced by many factors, including phase structure, particle morphology, crystallite size, specific surface area, crystal composition, crystallinity, and crystal facets [35,51]. Generally speaking, the specific surface area increases with a decrease in the particle size. Appendix A shows the particle size distributions of pH2.5-T175, pH4.5-T175, pH6.5-T175, and P25 measured from the enlarged photograph of FESEM images, and the corresponding parameters are listed in Table 1. Since degradation occurs on the surface of catalysts, the surface area is an important parameter for photocatalytic activity. The specific surface area was ranked in the order of pH2.5-T175 > pH4.5-T175 > P25 > pH6.5-T175, which was opposite to the increasing order of the average particle size (Table 1). The photocatalytic activity increased in the order of pH2.5-T175 > pH4.5-T175 > P25 > pH6.5-T175 > blank. The pH2.5-T175 sample exhibited the highest photocatalytic activity of the samples, while pH6.5-T175 showed the lowest photocatalytic activity of the samples. On the basis of the results, the order of the photocatalytic activity can be explained by considering the following factors: (1) A smaller particle size can provide a powerful redox capability in the photochemical reaction due to the quantum size effect, resulting in an increase in the rate of migration and a decrease in the rate of recombination of the photoelectrons and photogenerated holes, thereby improving the photocatalytic activity [52]. (2) The larger specific surface can provide more adsorption sites, resulting in an enhancement of MO adsorbed on the surface of TiO_2_, thereby contributing to the improvement of the photocatalytic activity [53].

In order to better understand the intrinsic photocatalytic activity of the different TiO_2_ nanocrystals, we examined the degradation amount of MO per unit surface area of catalyst (mg (MO) per m^2^ (TiO_2_ surface area)), as shown in Figure 12d. The degradation amount of pH2.5-T175, pH6.5-T175, P25, and pH4.5-T175 was calculated to be 0.13, 0.25, 0.27, and 0.31 mg/m^2^, respectively. On the basis of the above results and discussions, we know that the co-exposed facets of pH2.5-T175, pH4.5-T175, pH6.5-T175, and P25 were {101}/[111] facets, {101}/[111] facets, {010}/{001} facets, and {101}/[111] facets [54], respectively. Although the exposed crystal facets of the pH2.5-T175 and pH4.5-T175 nanocrystals were the same, their photocatalytic performances were quite different, indicating that the crystal facets were not the main factor affecting the photocatalytic performance. The degradation amount of MO per unit surface area of the pH2.5-T175 nanocrystals was the smallest among the samples, which can be attributed to them having the lowest crystallinity. Except for pH2.5-T175, theother samples all had high crystallinity, so their photocatalytic activity depended on the crystal plane. For the low-index facets of anatase, the average surface energy (γ) can be arranged in the following order: γ_{111}_ (1.61 J/m^2^) > γ_{110}_ (1.09 J/m^2^) > γ_{001}_ (0.90 J/m^2^) > γ_{010}_ (0.53 J/m^2^) > γ_{101}_ (0.44 J/m^2^) [55]. The P25 sample contained~87% anatase nanocrystals (partially co-exposed in the {101}/[111]-facets) and ~13% rutile nanocrystals, with an average size of ~26.2 nm [54]. The degradation amounts of MO per unit surface area of the pH4.5-T175 and P25 nanocrystals were higher than that of the pH6.5-T175 nanocrystal, which can be attributed to the existence of oxygen vacancies (active reaction sites) on the [111] facets of the TiO_2_ nanocrystals [14]. The pH4.5-T175 nanocrystal exhibited higher degradation amounts than did the P25 nanocrystal due to the larger proportion of [111] facets exposed on the surface of TiO_2_, which could create more active reaction sites in the process of photocatalytic reaction.

### 3.6. Photovoltic Performances of As-Synthesized TiO_2_ Nanocrystals 

The photocurrent–voltage characteristics of the DSSCs with the as-synthesized TiO_2_ nanocrystals and commercial Degussa P25 nanoparticles in the photoelectrodes are shown in Figure 13, and the detailed photovoltaic parameters of the four DSSCs, including open-circuit (*V*_OC_) and short-circuit photocurrent densities (*J*_SC_), the fill factor (*FF*), and the overall conversion efficiency (*η*), are summarized in Table 2. For comparison, the thicknesses of the TiO_2_ porous films were kept almost the same, at about 20 μm. The value of *J*_SC_ increased in the order of pH6.5-T175 < P25 < pH4.5-T175 < pH2.5-T175, which agreed with the decreasing order of the crystal size, while the value of *V*_OC_ increased in the order of P25 < pH2.5-T175 < pH4.5-T175 < pH6.5-T175, which was almost consistent with the increasing order of the crystal size, except for the P25 sample. The value of *FF* increased in the order of P25 < pH4.5-T175 < pH2.5-T175 < pH6.5-T175, which was almost the same as the *V*_OC_ increasing order, except for the pH2.5-T175 sample. The value of *η* increased in the same order as the *J*_SC_ value, i.e., the cell made of pH2.5-T175 nanocrystals showed the highest performance for the DSSCs, which represented an enhancement 20.7%, 25.1%, and 33.5% compared to the pH4.5-T175, P25, and pH6.5-T175 samples, respectively. It is well known that the particle size, specific surface area, film thickness, exposed crystal facets, crystal structure, and particle morphology of photoanodes are essential factors for enhancing photovoltaic properties [56,57]. The photoanode films of DSSCs that are prepared by smaller nanoparticles often possess a large internal surface area, which is beneficial to the adsorption of dye, resulting in an increase in the photocurrent and energy conversion efficiency [57]. The cells with the pH6.5-T175 electrode exhibited the lowest value of *J*_SC_ and *η* (Table 2) due to having the largest particle size and the lowest specific area (Table 1), which reduced the available internal surface area for the adsorption of the dye molecules, resulting in a decrease in light-harvesting efficiency [58]. The cells made of pH2.5-T175 possessed the highest *J*_SC_ and *η* values, which can be attributed to their minimum particle size and maximum surface area. 

In accordance with the previous discussion, the exposed crystal facets of the pH2.5-T175, pH4.5-T175, pH6.5-T175, and P25 nanocrystals were {101}/[111] facets, {101}/[111] facets, {101}/{010} facets, and {101}/[111] facets (only a small fraction), respectively. It has been reported that the adsorption equilibrium constant and the surface uptake density of N719 dye molecules on different crystal facets of the TiO_2_ surface increase in the order of the following: without specific exposed facets < {101} facets < [111] facets < {010} facets on the surface of anatase TiO_2_. In other words, the strong adsorption and surface uptake density of N719 dye molecules on the surface of anatase TiO_2_ can increase the light-harvesting efficiency, resulting in an improvement in the photovoltaic performance [59,60]. However, the {101}/{010} facets co-exposed in the pH6.5-T175 photoanode exhibited the lowest *J*_SC_ value, which implies that compared to the specific area, the crystal facets were not the most important effecting factor in this research. The cells with a pH6.5-T175 photoanode showed the highest *V*_OC_ and *FF* values, which may be attributed to the fact that the high surface uptake density can decrease the charge recombination at the TiO_2_/electrolyte interface [56], and the oriented anatase TiO_2_ nanospindle structure can improve the charge transport properties in the porous TiO_2_ film [57]. The photoanode made of pH2.5-T175 nanocrystals (co-exposed {101} and [111] facets) possessed a higher *J*_SC_ value than did the one made of pH4.5-T175 nanocrystals (co-exposed {101} and [111] facets) or P25 nanocrystals (partial co-exposed {101} and [111] facets) at a similar film thickness, which can be attributed to the higher specific surface area enhancing N719 dye adsorption, resulting in an increase in the *J*_SC_ value.

## 4. Conclusions

Microwave-assisted hydrothermal synthesis is an effective synthesis route to produce TiO_2_ nanocrystals with different morphologies and high-energy surfaces at relatively low temperatures, which can shorten the reaction time, reduce the energy consumption, and enhance the purity and crystallinity of the produced products compared to the traditional hydrothermal method. Titanate nanosheets were exfoliated from layered porous metatitanic acid with a lepidocrocite-type structure, which was suitable as a building block for the assembly of different structured TiO_2_ nanocrystals. The pH values of the nanosheets had a significant effect on the crystallinity, crystallite, phase structure, morphology, and exposure facets of the prepared TiO_2_ nanocrystals. Rutile TiO_2_ quadrangular prisms with dominant {110} facets, anatase TiO_2_ nanorods and square nanoprisms with co-exposed {101}/[111] facets, anatase TiO_2_ nanorhombuses with co-exposed {101}/{010} facets, and anatase TiO_2_ nanospindles with dominant {010} facets were synthesized through a facile green approach with the use of exfoliated porous metatitanic acid nanosheets as the precursor at various pH values. The morphology and exposed crystal facets of the obtained TiO_2_ nanocrystals could be controlled by adjusting the pH value of the nanosheet solution. The transformation reaction mechanism from the exfoliated metatitanic nanosheets with distorted hexagonal cavities to TiO_2_ nanocrystals could include a dissolution reaction of the nanosheets, nucleation of the primary [TiO_6_]^8−^ monomers, and the growth of rutile-type and anatase-type TiO_2_ nuclei. The {101}/[111]-faceted pH2.5-T175 nanocrystal showed the highest photocatalytic and photovoltaic performance compared to the other TiO_2_ samples, which could be attributed mainly to its minimum particle size and maximum specific surface area.

## Figures and Tables

**Figure 1 materials-12-03614-f001:**
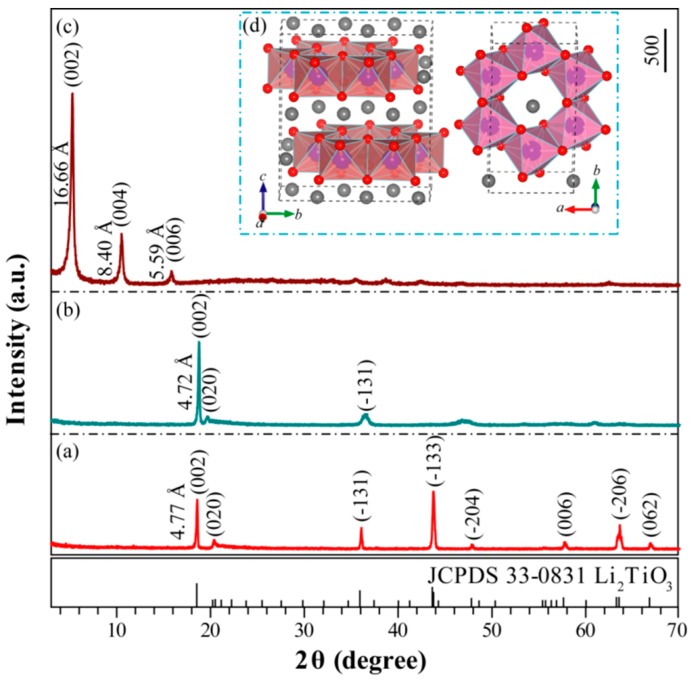
X-ray diffraction (XRD) patterns of (**a**) Li_2_TiO_3_, (**b**) H_2_TiO_3_, and (**c**) TMA^+^-intercalated H_2_TiO_3_ samples; and (**d**) structural model of Li_2_TiO_3_ viewed in the *a* axis and *c* axis directions.

**Figure 2 materials-12-03614-f002:**
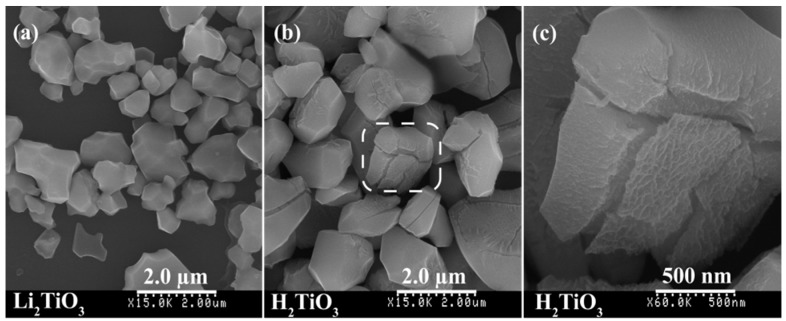
Field-emission (FE)SEM images of (**a**) Li_2_TiO_3_, and (**b****,c**) H_2_TiO_3_.

**Figure 3 materials-12-03614-f003:**
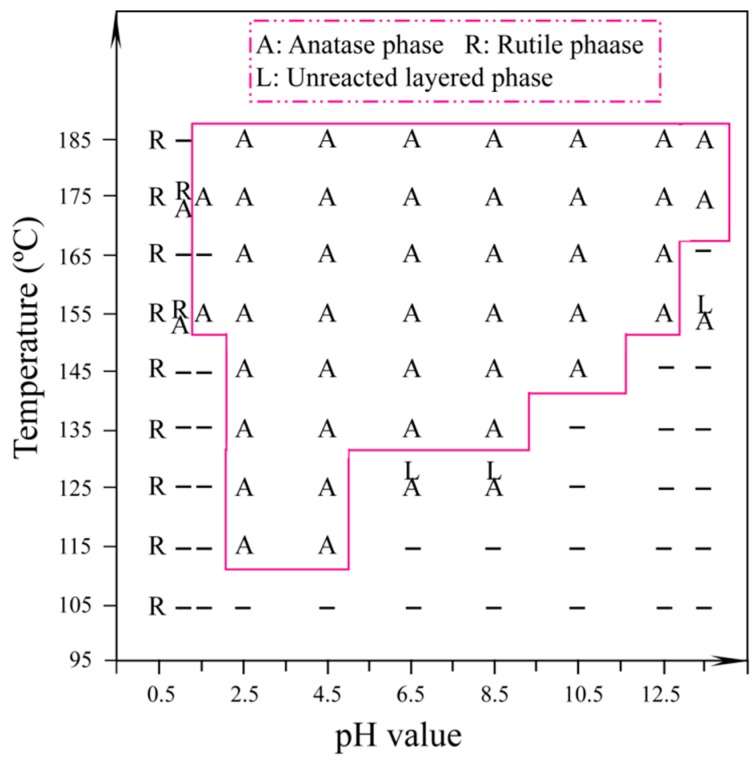
Changes in the crystal form of titanium dioxide with temperature and pH value.

**Figure 4 materials-12-03614-f004:**
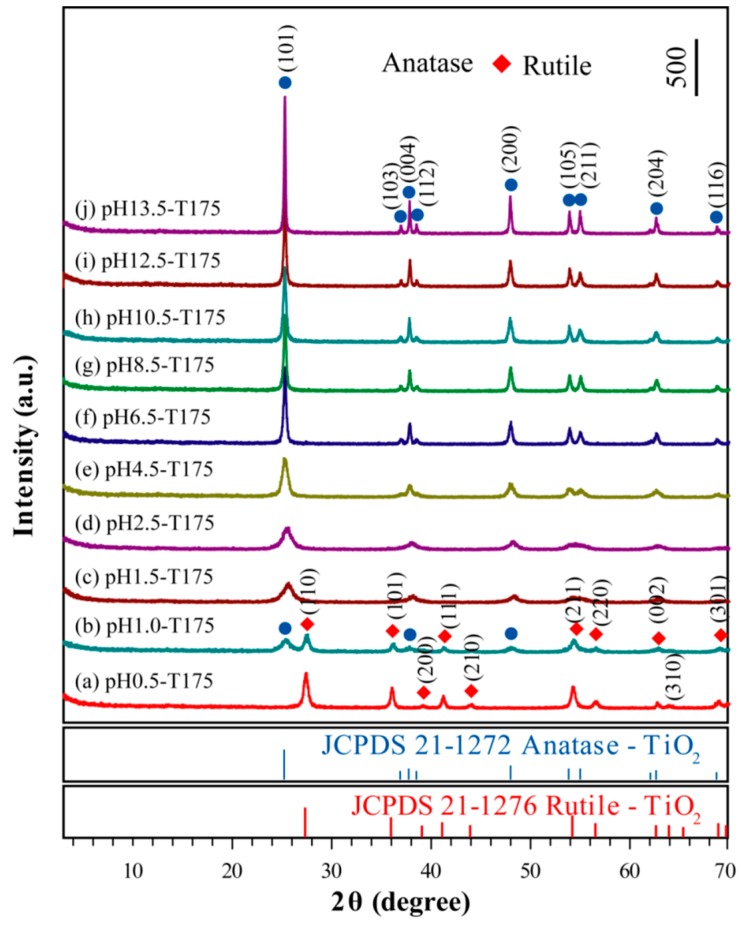
Evolution of the XRD patterns of the TiO_2_ nanocrystal samples synthesized at various pH values with the indicated lattice phases of (●) anatase and the (◆) rutile phases.

**Figure 5 materials-12-03614-f005:**
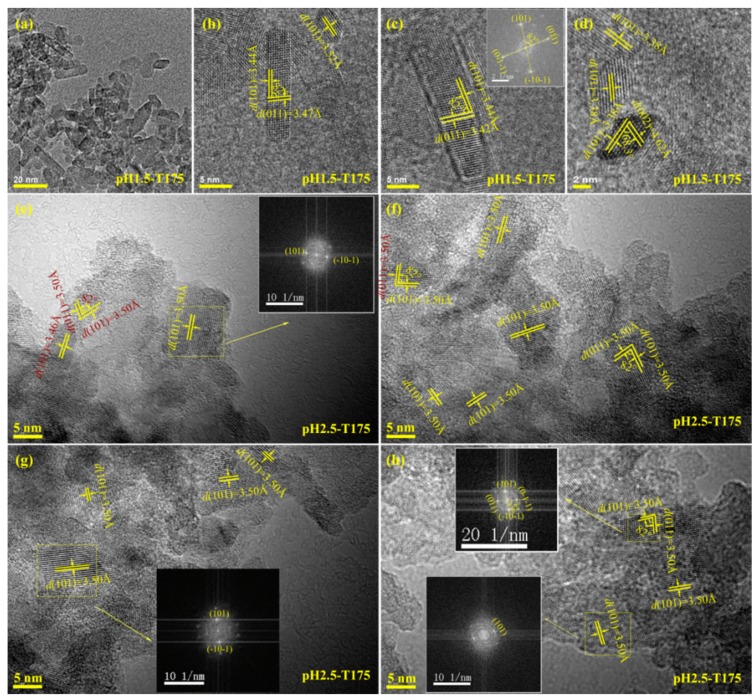
(**a**) TEM and (**b–d**) high-resolution (HR)TEM images of (**a–d**) pH1.5-T175 and (**e–h**) HRTEM images of pH2.5-T175 samples prepared by microwave-assisted hydrothermal treatment of the exfoliated H_2_TiO_3_ nanosheet colloidal solution at 175 °C with a reaction time of 2 h.The insets in (c,e,g,h) are fast-Fourier-transform (FFT) diffraction patterns.

**Figure 6 materials-12-03614-f006:**
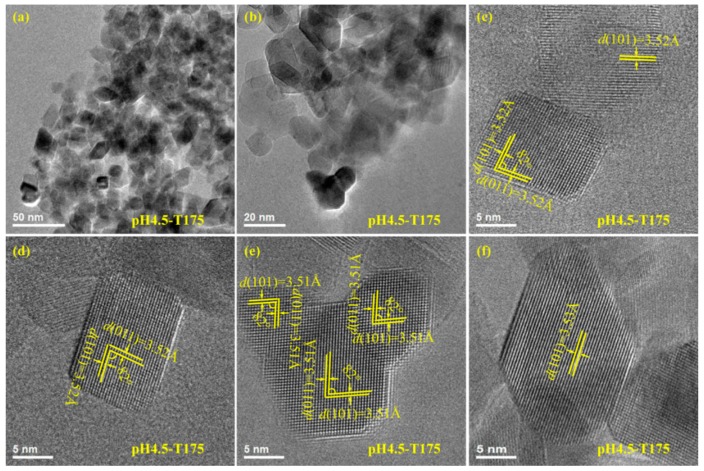
(**a,b**) TEM images and (**c–f**) corresponding HRTEM images of the pH4.5-T175 sample prepared by microwave-assisted hydrothermal treatment of the exfoliated H_2_TiO_3_ nanosheet colloidal solution at 175 °C with a reaction time of 2 h.

**Figure 7 materials-12-03614-f007:**
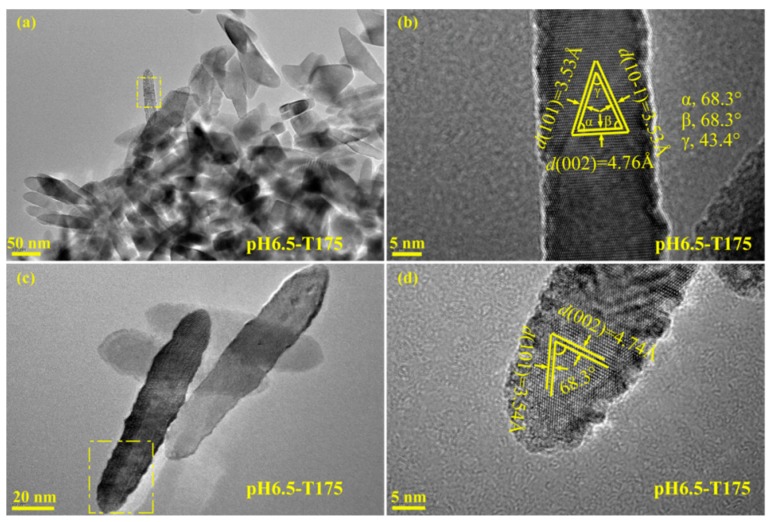
(**a,c**) TEM images and (**b,d**) corresponding HRTEM images of the pH6.5-T175 sample derived from microwave-assisted hydrothermal treatment of the exfoliated H_2_TiO_3_ nanosheet colloidal solution at 175 °C with a reaction time of 2 h.

**Figure 8 materials-12-03614-f008:**
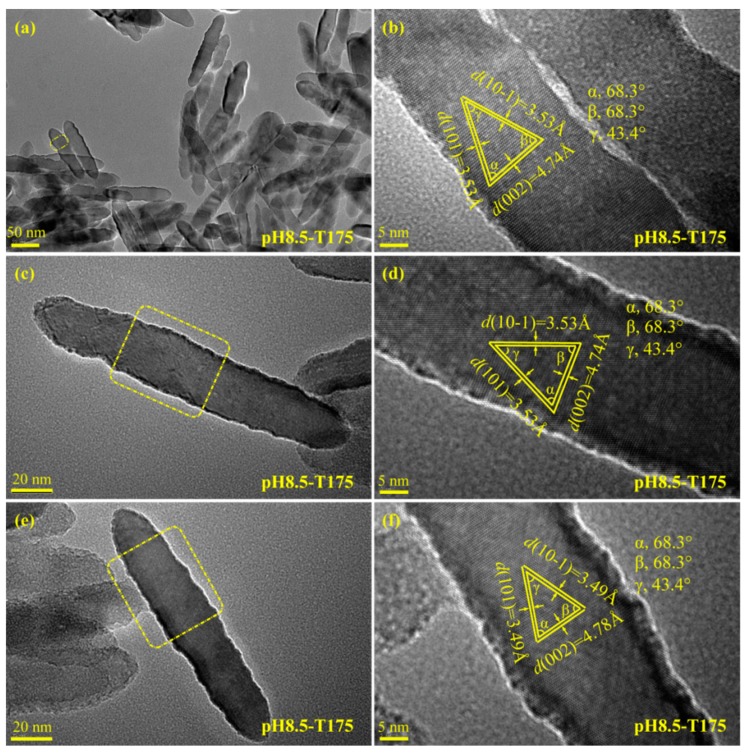
(**a,c,e**) TEM images and (**b,d,f**) corresponding HRTEM images of the pH8.5-T175 sample derived from microwave-assisted hydrothermal treatment of the exfoliated H_2_TiO_3_ nanosheet colloidal solution at 175 °C with a reaction time of 2 h.

**Figure 9 materials-12-03614-f009:**
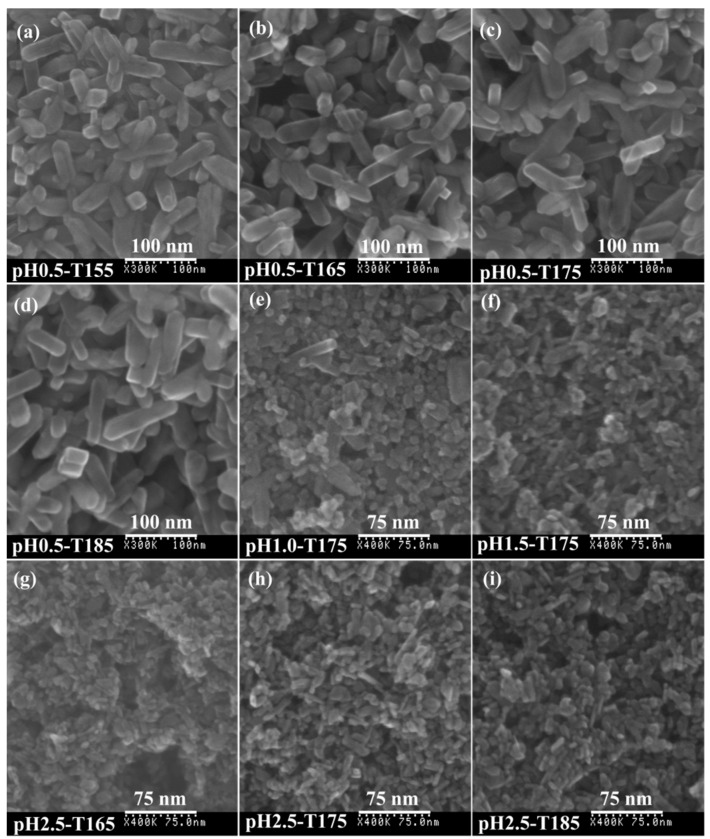
FESEM images of TiO_2_ nanocrystals synthesized under different pH values and temperatures: (**a**) pH0.5-T155, (**b**) pH0.5-T165, (**c**) pH0.5-T175, (**d**) pH0.5-T185, (**e**) pH1.0-T175, (**f**) pH1.5-T175, (**g**) pH2.5-T165, (**h**) pH2.5-T175, and (**i**) pH2.5-T185.

**Figure 10 materials-12-03614-f010:**
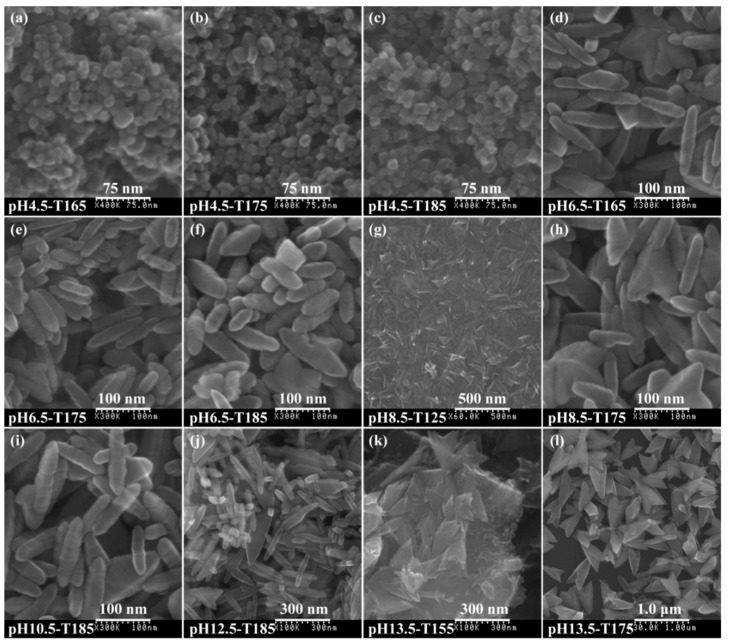
FESEM images of the TiO_2_ nanocrystals synthesized under different pH values and temperatures: (**a**) pH4.5-T165, (**b**) pH4.5-T175, (**c**) pH4.5-T185, (**d**) pH6.5-T165, (**e**) pH6.5-T175, (**f**) pH6.5-T185, (**g**) pH8.5-T125, (**h**) pH8.5-T175, (**i**) pH10.5-T185, (**j**) pH12.5-T185, (**k**) pH13.5-T155, and (**l**) pH13.5-T175.

**Figure 11 materials-12-03614-f011:**
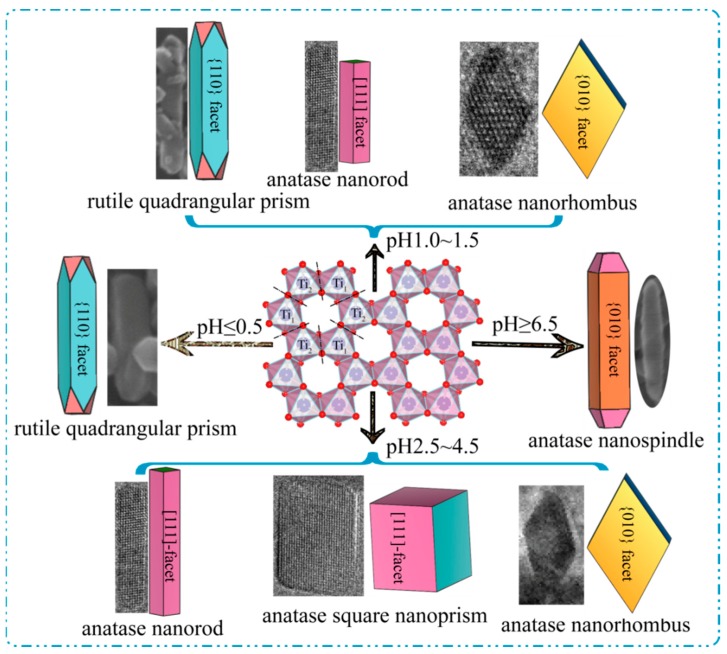
Schematic diagramof the formation mechanism of the TiO_2_ nanocrystals with different morphologies from the exfoliated H_2_TiO_3_ nanosheets at different pH values.

**Figure 12 materials-12-03614-f012:**
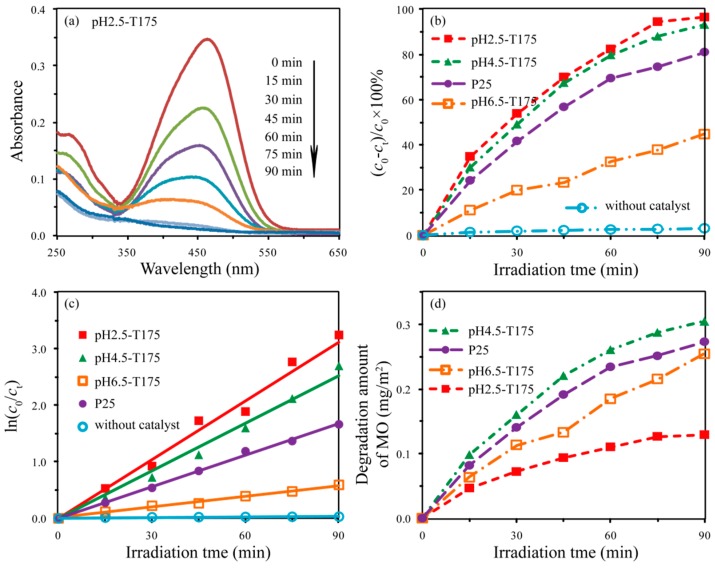
(**a**) UV-Vis spectral changes of the centrifuged methyl orange (MO) solutions at certain intervals during the photodegradation treatment with pH2.5-T180; (**b**) photocatalysis degradation profiles of MO under UV irradiation; (**c**) first-order kinetics fitting data for the photodegradation of the MO; (**d**) degradation amount of MO per unit area of TiO_2_ nanocrystals.

**Figure 13 materials-12-03614-f013:**
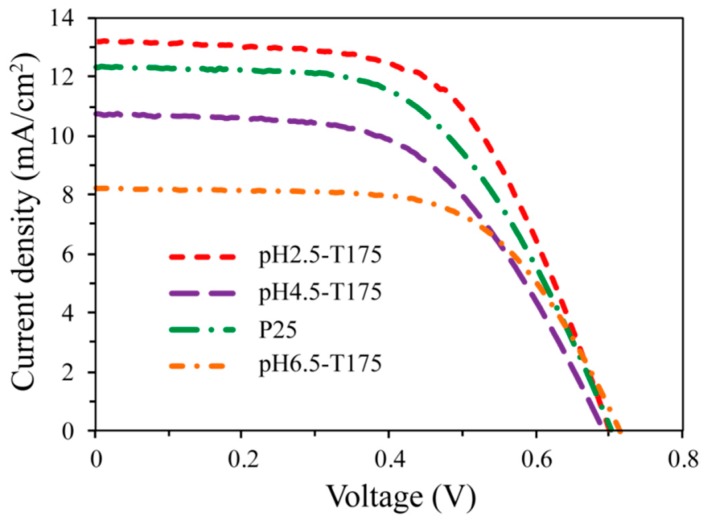
Current–voltage characteristic curves of dye-sensitized solar cells (DSSCs) fabricated using pH2.5-T175, pH4.5-T175, pH6.5-T175, and P25 samples.

**Table 1 materials-12-03614-t001:** specific surface area, average particle size (nm), *k*, and *R*^2^ values of the TiO_2_ nanocrystals.

Samples	Specific Surface Area (m^2^/g)	Average Particle Size (nm)	Degradation Rate Constant *k* (min^−1^)	Correlative Coefficient (*R*^2^)
pH2.5-T175	135.6	8.3	0.0346	0.9502
pH4.5-T175	54.8	13.3	0.0281	0.9846
pH6.5-T175	32.3	87.0	0.0065	0.9920
P25	52.5	23.9	0.0187	0.9971
blank			0.0004	0.9595

**Table 2 materials-12-03614-t002:** Cell performance parameters of DSSCs fabricated using different TiO_2_ samples.

Photoelectrode	Film Thickness (μm)	*J*_SC_ (mA/cm^2^)	*V*_OC_ (V)	*FF*	*η* (%)
pH2.5-T175	19.17	13.19	0.701	0.595	5.50
pH4.5-T175	20.03	12.32	0.704	0.556	4.83
pH6.5-T175	19.11	8.22	0.715	0.622	3.66
P25	20.40	10.73	0.693	0.554	4.12

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
