# Peer review of "Microwave-Assisted Synthesis of High-Energy Faceted TiO2 Nanocrystals Derived from Exfoliated Porous Metatitanic Acid Nanosheets with Improved Photocatalytic and Photovoltaic Performance"

_materials, 2019, doi:10.3390/ma12213614_

Round 1

Reviewer 1 Report

The manuscript shows a facile one-pot microwave-assisted hydrothermal synthesis of rutile TiO2 quadrangular prisms with dominant {110} facets, anatase TiO2 nanorods and nanocuboids with coexposed {101}/[111]-facets, anatase TiO2 nanorhobus with coexposed {101}/{010} facets, and anatase TiO2 20 nanospindles with dominant {010} facets were reported through using the exfoliated porous 21 metatitanic acid nanosheets as the precursor. It is well presented however, several issues must be addressed.  

English errors

Therefore, it is imperative to reasonably design and synthesize various TiO2 68 nanocrystals with high energy surfaces exposed for improving the photocatalytic activity and 69 optimizing the DSSC performance has attracted great interest and exploration. The above suspension solution was placed in the dark for 2 days for 2 days to ensure that adsorption-desorption phenomenon had reached equilibrium. In can been seen that rutile phase is formed preferentially under strong 246 acidic conditions (pH ≤ 0.5). The {101}/[111]-faceted pH2.5-T175 nanocrystal showed the higher photocatlaytic and 628 photovoltaic performances, compared to those of other TiO2 samples, which can be attributed mainly to the smaller particle size and the higher specific surface area.   More comments:   It has not been said the advantages of the microwave synthesis or the production route used.   It has not been justified why there is a mixture of phases after microwave synthesis. It is just said that it is a effect of pH, but why? How the pH influences the shape of the nanostructures?   The FFT images for the TEM measurements should be presented and indexed to support the findings. In fact it is really hard to see the measurements on the TEM images.   It has been said:  Moreover, a small number of nanocrystals with irregular morphologies were also observed in the above sample. Is it another phase? How can you prove it. One example is the paper: Photocatalytic TiO2 nanorod spheres and arrays compatible with flexible applications. It has been shown the presence of different nanoparticles and it has been shown the TiO2 phase. Please add the reference and discuss with your results.   The photocatalytic behaviour is not extraordinary but acceptable. Is it possible to perfom photocatalysis using white light or solar light? It is important to guarantee the materials flexibility to general use. The conclusion is short and not reveals the novelty of the study. The references does not reveal an extensive research or diversity and it is mostly authors from Asia. I suggest a better look on literature.   Best regards  

Reviewer 2 Report

The authors report formation of different TiO2 nanostructures through a microwave-assisted process. Although the current research is well studied, the manuscript lacks novelty as it is very similar to previous papers from the same group. What is the difference between the present work and previous articles (reference 9, 23, 24, 52 and ChemistrySelect 2018, 3, 9953 –9959)? These articles are very similar in the synthesis, resulted morphologies and functionality applications in which P25 is slightly less active than the produced photocatalyst. It is mentioned in the introduction that anatase shows the best performance in comparison with other polymorphs. However, the authors compared their pure anatase samples with P25 (a mixture of anatase and rutile). Therefore, I cannot recommend publication.

Reviewer 3 Report

Comments:

In this manuscript entitled “Microwave-assisted synthesis of high-energy faceted TiO2 nanocrystals derived from the exfoliated porous metatitanic acid nanosheets with improved photocatalytic and photovoltaic performances ”, the authors claimed that the microwave-assisted, a low price and nontoxicity method, with manipulation the PH value can be used to synthesize the TiO2 for photovoltaic device application. Authors used a lots of techniques to characterization the TiO2 crystal structures and photovoltaic performances. However, the organization of the manuscript may confuse the readers and make the arguments and analyses more difficult to follow. The reorganization of the manuscript can help readers easier to understand and follow the authors’ arguments.     

Authors may put the figure 9 schematic and the formation of the different crystal structure of TiO2 at various reaction conditions (PH value and temperature) before the electron microscopy analysis. It can help reader to understand the overall concept of this manuscript.

Authors analyzed the SEM images and claim the crystallinity growth facets and growth direction without any explanation. It is very hard to follow the authors’ arguments without showing the HRTEM images. So it is better to analyze HRTEM images first, and then discussion the crystallinity growth direction by SEM images.    

In page 8 and 9, authors claim that “increase of temperature only increased the crystallinity of nanoparticle.” without any proof. Author need to add the XRD spectra analysis to prove their claim.

Authors claim that the morphology of TiO2 at PH=2.5~4.5 is rod-shaped, cuboid-shaped and some irregular shape. Based on the TEM images, it is very hard to make the conclusion of cuboid-shaped. Authors need to clarify the morphology is cuboid-shaped or square prism-shaped nanoparticles

Authors need to check the manuscript again for correction some errors There are no figure 2(b) which authors mention in page 7. Page 11 the sample is PH 4.5-T175 not PH 15-T175 Page 16 : P25(81.1%)>PH6.5-T175(44.7%)>blank sample (3.1%); not P25(81.1%)>PH4.5-T175(44.7%)>blank sample (3.1%);

Round 2

Reviewer 1 Report

The manuscript is very well presented in the current form.

Reviewer 2 Report

The authors has replied to my concerns.